# Systematic analysis of noise reduction properties of coupled and isolated feed-forward loops

**Suchana Chakravarty**[1]*, **Attila Csikász-Nagy**[1,2]*

**1** Faculty of Information Technology and Bionics, Pázmány Péter Catholic University, Budapest, Hungary,
**2** Randall Center for Cell and Molecular Biophysics, King's College London, London, United Kingdom

* chakravarty.suchana@itk.ppke.hu (SC); csikasz-nagy.attila@itk.ppke.hu (AC-N)

## Abstract

Cells can maintain their homeostasis in a noisy environment since their signaling pathways can filter out noise somehow. Several network motifs have been proposed for biological noise filtering and, among these, feed-forward loops have received special attention. Specific feed-forward loops show noise reducing capabilities, but we notice that this feature comes together with a reduced signal transducing performance. In posttranslational signaling pathways feed-forward loops do not function in isolation, rather they are coupled with other motifs to serve a more complex function. Feed-forward loops are often coupled to other feed-forward loops, which could affect their noise-reducing capabilities. Here we systematically study all feed-forward loop motifs and all their pairwise coupled systems with activation-inactivation kinetics to identify which networks are capable of good noise reduction, while keeping their signal transducing performance. Our analysis shows that coupled feed-forward loops can provide better noise reduction and, at the same time, can increase the signal transduction of the system. The coupling of two coherent 1 or one coherent 1 and one incoherent 4 feed-forward loops can give the best performance in both of these measures.

## Author summary

Cellular behavior can be affected by noise in molecular interactions. Signaling pathways should process noisy input signals and support cellular decision making by properly transducing the signals, while removing noise from them. Three component networks of feed-forward loops (FFLs) have been proposed to serve as ideal noise reducers, while linear pathways were shown to be good signal transducers. These signaling units do not work in isolation, so there is a possibility that a combination of various feed-forward loops can provide good noise reduction, while maintaining good signal transduction. To test this hypothesis, we have systematically tested the noise reducing and signal transducing capabilities of all possible combinations of feed-forward loops and compared them with the performance of individual FFLs. We built mathematical models of all these systems and compared their capabilities at reducing noise in the input signal while maintaining

---

**Data Availability Statement:** The codes are uploaded in Github https://github.com/SuchanaChakravarty/Noise-Reduction-Properties-of-Coupled-and-isolated-Feed-Forward-Loops The

rest of the relevant data is within the manuscript and its Supporting Information files.

**Funding:** This work was supported by the National Research, Development and Innovation Office of Hungary (K_20 134489) and (Thematic Excellence Programme - TKP2020-NKA-11) (to AC-N). The funders had no role in study design, data collection and analysis, decision to publish, or preparation of the manuscript.

**Competing interests:** The authors have declared that no competing interests exist.

responses to meaningful changes in the incoming signal. We found that a combination of two copies of a special type of fully positive signaling FFLs is the best noise reducer, while a combination of two incoherent (one positive, one negative signal) FFLs can provide the best signal transduction. The combination of these two FFLs could provide good signal processing where both noise reduction and signal transduction are achieved.

## Introduction

Random fluctuations in molecular levels causes noise in various biological processes [1–3]. Cells must make crucial decisions in such noisy molecular environment thus their decision-making system should be able to control stochastic fluctuations. Extrinsic noises can be caused by the noisy environment around cells [4], whereas low-copy number molecules generate random fluctuations, which we term intrinsic noise [5–7]. This intrinsic noise is directly proportional to the square root of the number of molecules presents in system [8]. The resultant fluctuations in molecule numbers might result in increased noise in processes controlled by low-abundance molecules. Gene expression is a typical example, where a few copies of transcription factors influence the transcription [9] of several genes.

Many people have so far examined how this noise does not affect the homeostatic behavior of cells. Special wiring in the molecular interaction network was suggested to minimize noise. In order to illustrate how a biological network may operate as a low-pass filter, engineering ideas were transferred into biology. Typical network motifs that might be used as noise filters were identified. Special interest was drawn among these to feed-forward loop (FFL) motifs [10]. In a feed-forward loop, a molecule both directly and indirectly affects the activity of a downstream target [11]. The conventional depiction of an FFL comprises of 3 nodes, where X directly regulates Y and Z, and Y also affects Z, leading to an extra indirect effect between X and Z (S1A Fig). In our illustrations, we also include a signaling molecule S, which provides the initial input to X (Fig 1A). If the direct arm connecting X and Z has the same sign as the indirect arm (through Y), we have a coherent feed-forward loop (cFFL). If the two arms have the opposite signs, then it is called an incoherent feed-forward loop (iFFL). A type-1 cFFL (c1FFL) contains only activation steps and it was identified as the most common three component network motif in several biological systems. It was also identified that c1FFL can work as a noise reducing low-pass filter [12,13].

Noise attenuation can be also achieved by a negative feedback loop, which is also widely observed in biological regulator networks [14–16]. Such noise filtering capabilities are not restricted to the regulation of mRNA levels [17], but signaling pathways should also be able to filter out noise when a signal is received through the activation of membrane receptors. The linear pathway of three mitogen activated protein kinases (MAPK) is also capable of noise reduction [18]. Noise reduction was further explored in posttranslational regulatory networks, and many additional motifs were found as noise filters. The annihilation module has two molecules with correlated production and combined degradation (Fig 1B), which serves as an efficient noise reduction motif but imposes a delay on the system. The combination of the annihilation module with an iFFL resulted in a very effective noise-reducing annihilation filter that does not even add a delay in the system [19]. Indeed, these network motifs usually cannot be observed in isolation, but they are coupled to each other. For instance, Gershom Bur *et al.* showed that coupled coherent and incoherent feed-forward loops regulate immune cells [20]. Another example is the dense crosstalk between MAPK pathways. Coupling of specific feed-forward loops was also found in other systems [21]. Such coupled FFLs have double-input and

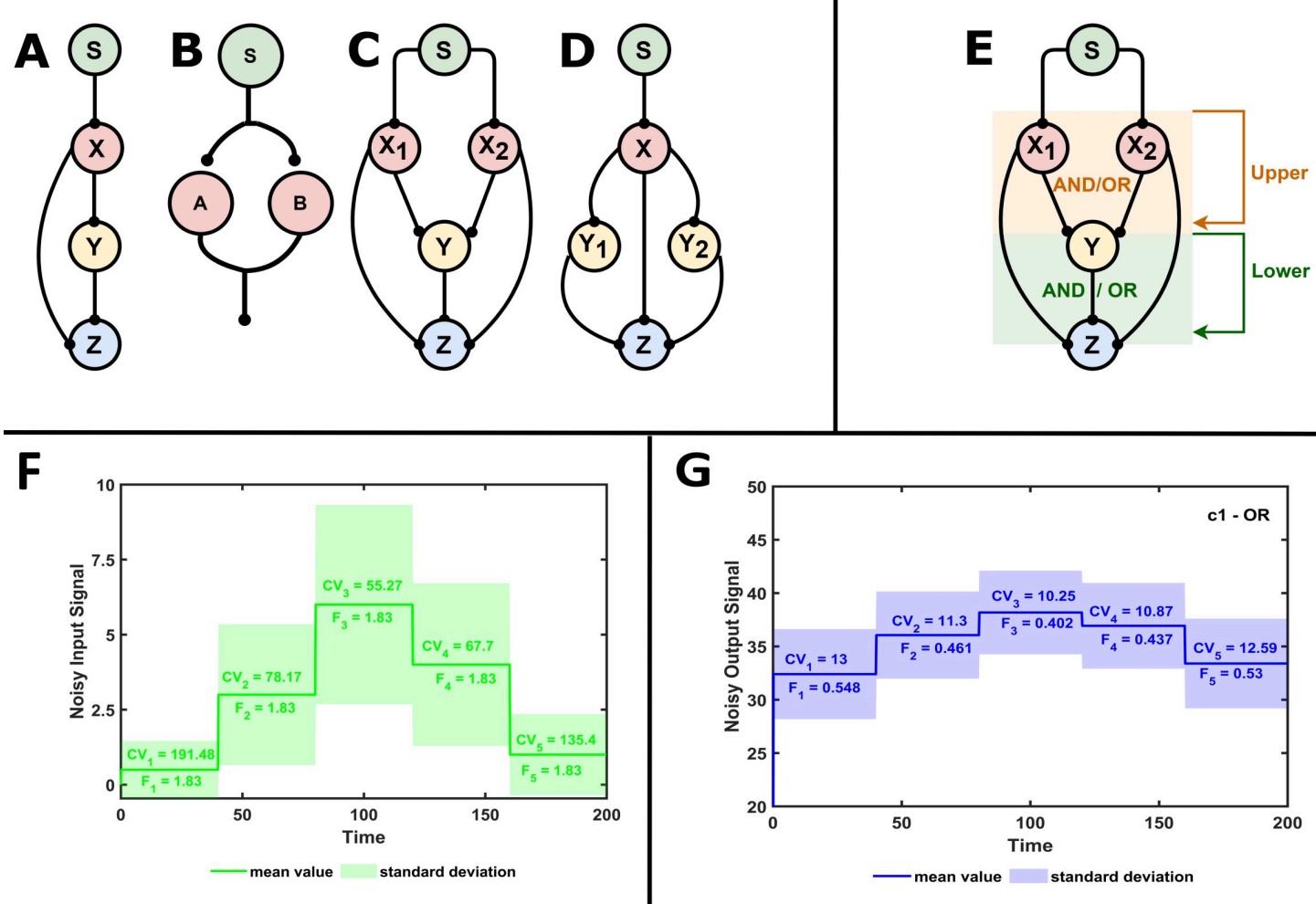

**Fig 1. Network motifs and the input-output test ran on them.** Feed-forward loop (A), annihilation module (B), combination of feed-forward loops, which contain traces of the annihilation module: multi-input coupled feed-forward loop (minp-FFL) (C) and multi-intermediate coupled feed-forward loop (mint-FFL) (D)–pointed-end arrows can stand for activating or inhibiting interactions. AND or OR logical gates are considered for all FFL models, but in minp-FFL logic gates function at two separate levels leading to 4 possible systems: fully AND, fully OR, upper-AND-lower-OR (uAND-lOR) and upper-OR-lower-AND (uOR-lAND) (E). In all models, the input signal (*S*) was driven by super-Poissonian noise (the coefficient of variation in percentage (% CV) at all average input values is presented and the resulting Fano factor of 1.83 at all average input levels is also noted.) (F). The average output level and the noise of the output species *Z* was registered for all input levels, leading to reduced Fano factor and coefficient of variation (% CV) values, which still depend on the average input level (G).

double-output reactions (Fig 1C and 1D), similar to the ones observed in the annihilation module (Fig 1B). This raises the possibility that such coupled FFLs could be quite efficient in noise reduction.

A very effective noise reduction system would eliminate the effects of any input on the system, but lose the capacity for the signal transduction [22,23]. A signal processing network should reduce the noise affecting the cells but should be able to respond to meaningful changes [24]. Low-pass filters can nicely deal with this, but a perfect implementation of these in biological systems is still lacking. Here we test systematically all possible combinations of FFLs and compare their noise reducing and signal transducing capabilities to the performance of single FFLs and linear three-layered pathways with matching parameter sets. For this analysis we consider posttranslational modifications activating and inactivating molecules at all layers of the FFLs.

## Results

Coupled feed-forward loops can be categorized as multi-input coupled feed-forward loops (minp-FFL) (Fig 1C) and multi-intermediate coupled feed-forward loops (mint-FFL) (Fig 1D). Depending on the signs of the individual interactions, the coupled FFLs could be fully coherent, fully incoherent or a mixture of coherent and incoherent. 47 possible FFL and combined FFL networks were identified (S1 Fig). If several molecules impact the component downstream, the two input signals can influence the objective separately (OR logic gate) or together (AND logic gate). In the case of multi-input coupled feed-forward loops (minp-FFL), such logic gates appear at two levels in the system, introducing four possible combinations: fully AND, fully OR, upper-AND-lower-OR (uAND-lOR) and upper-OR-lower-AND (uOR-lAND) (Fig 1E). When the incoming signals have similar sign effects on the target molecule, both OR and AND gates can be considered; however, when the regulations have opposite signs, only one gate type can be considered, which is not fully a classical OR or AND gate, but it is closer to an OR gate, so we will use this notation in the text. In the case of minp-FFLs, other combinations also drop out. With these restrictions, we ended up investigating 59 models of possible FFL combinations to compare their behavior to 12 FFLs and 4 possible linear chain models where only a single positive or negative effect runs through a three-component system (S1 Fig).

Since we focus on noise reduction in signaling pathways, we consider activation and inactivation steps as post-translational modifications. Post-translational modifications, such as phosphorylation, often happen at multiple sites on target molecules. We consider two versions of all models. In *one-step modification models*, there is a single mass-action conversion between active and inactive forms, while in *two-step modification models*, the transition happens through an intermediate. Such two-step processes give similar dynamics to back-and-forth enzymatic reactions, also termed as Goldbeter-Koshland switches [25–27]. In the paper we focus on single-step modification models. The two-step modification models are presented in the supplement and only discussed in the main text. At each layer of the regulation, we consider the total abundance of molecules to be 60 arbitrary unit. The parameters of each system were set to the same values at each layer, only halving the effects of regulators when functioning in an AND gate, to keep the average output the same at an input value of 1. The parameters had to be chosen in a way that neither of the network models would end up saturating the activation of all molecules, nor should they inactivate all molecules either. Coherent activating and coherent inhibiting systems were also considered, and they move output in opposite directions, thus the chosen parameter set restricted the input-output relationship and the noise reducing capabilities of the best networks. The exact choice of these parameters to ensure the total molecule number never gets saturated, and the selected values are given in the supplementary text (S1 Text).

Stochastic dynamical equations for any Chemical Reaction Networks (CRNs) can be solved by the Chemical Master Equation (CME) [28]. But solving the CME could be challenging and with large molecule numbers, almost impossible. Analysis of the stochastic nature of CRNs can be done by either rigorous continuous-time Markov chain (CTMC) [29] or Gillespie Simulation [30]. An intermediate solution is the use of linear noise approximation (LNA) [31]. Linear noise approximation is a fast approximation method for CRNs [32]. We ran LNA on our models to calculate the distribution of each species by coding the models in the recently released 'Kaemika' tool [33]. In the supplementary text (S1 Text) we provide annotated examples of the Kaemika codes, explaining how to create each of the presented models.

To test the noise reducing and signal transducing capabilities of the investigated systems, we implemented a five-step increase and decrease in the input signal 'S' in our models (Fig 1F)

and followed how the output layer 'Z' responds to these changes (Fig 1G). By introducing a bimolecular production term, a super-Poissonian [34] noise was generated on the input $S$ molecule and the noise in the output layer was registered. Interestingly we did not notice major delays in output responses as the mean input was changed in an up-and-down fashion (Fig 1F and 1G). To quantify noise, we have calculated the coefficient of variation in percentage (% CV) for all molecule types. Fano factor measures noisiness of the signal, but CV is a better measure of signal-to-noise ratio, which is relevant in signal transduction, thus we report these values on plots. Coefficient of variation is the ratio of the standard deviation and the mean of the random process [35, 36]. Poissonian noise has a coefficient of variation of (mean value)$^{-1/2}$ and a Fano factor of 1. A lower coefficient of variation (or Fano factor) indicates greater noise reduction. We quantified the noise reducing capabilities of the networks by calculating the slope of mean output–mean input relation, where we plot these values for the five different input levels (Fig 2A and 2B). We use this 'slope' on plots as a measure of signal transduction capacity, with higher slopes meaning better signal transduction, while negative slopes mean signal inversion.

## Comparison of simple feed-forward loops

Both coherent and incoherent FFLs can give positive or negative slopes, depending on the signs of each individual regulatory step (Fig 2A and 2B). Because all of the outputs have a lower coefficient of variation than the input coefficient of variation value, each of these FFLs can reduce noise (Fig 2C). Larger input results in lower coefficient of variation, since the average molecule numbers also increase with larger input, leading to reduced noise. At these larger

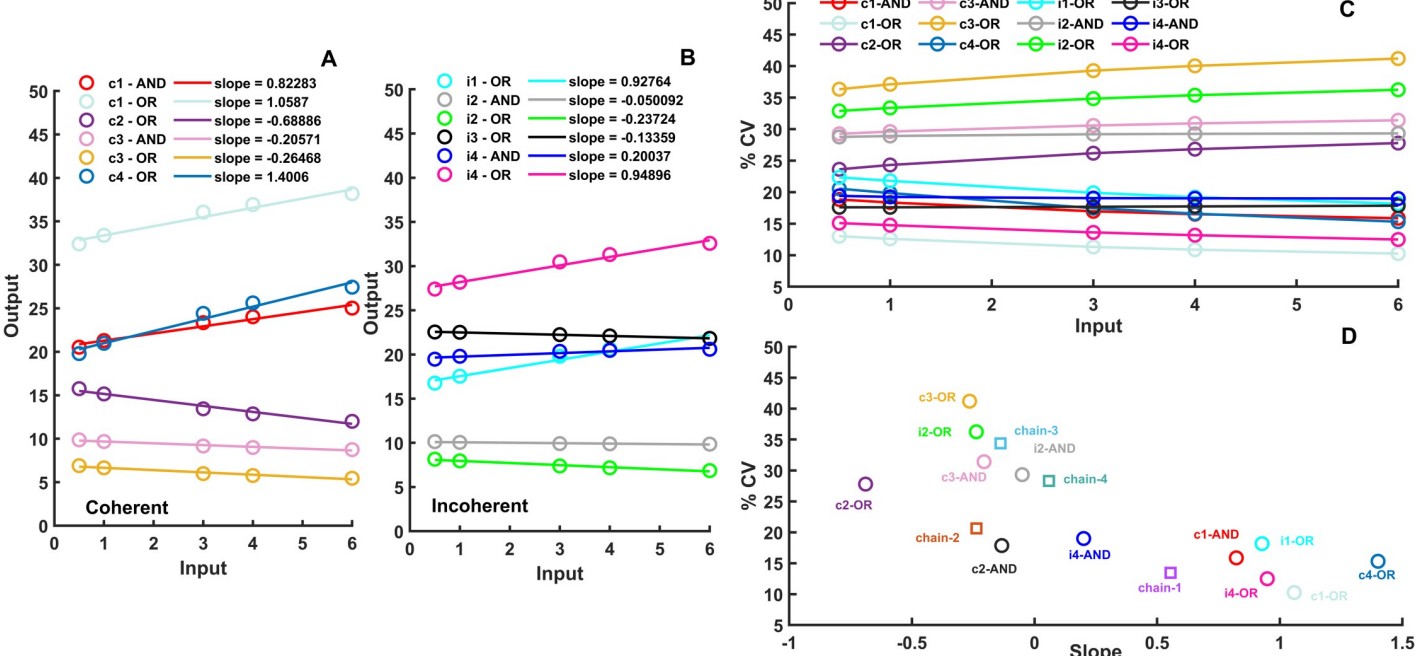

**Fig 2. Signal transducing and noise reducing capabilities of linear chain and simple feed-forward loop network motifs.** Mean input–mean output relation in the investigated post-translational coherent (A) and incoherent (B) FFLs observed at 5 different input levels. The noise reducing capabilities—measured by the coefficient of variation in percentage (%CV) at all tested input values in all FFLs (C). The coefficient of variations in percentage (%CV) of panel C at input = 6 is plotted against the slopes of the input-output relations of panels A and B to show how each network performs in noise reduction and signal transduction (D). The performance of linear chain motifs (S6 Fig) is also added to panel D. All simulations were run with one-step post-translational modification models and the fixed parameter sets presented in the S1 Text.

input values, the FFLs show larger difference. We used results from the simulations with input = 6 and plotted %CVs against the slopes from (Fig 2A and 2B) to demonstrate how FFLs compare in terms of noise reduction and signal transduction (Fig 2D). Interestingly better noise reduction (lower coefficient of variation) correlates with better signal transduction (larger slope). The c1- OR network performs the best in both measures, what is not surprising as it was shown to be able to work as a low-pass filter [37] and also as a good signal transducer [23]. As reference points, we have also plotted the coefficient of variation and slope produced by linear chain models (Fig 2D). We see that some noise reduction can be achieved by such a simple signaling cascade, due to the nonlinearity caused by the presence of a limited number of total molecules, which are converted between active and inactive forms. 3 out of 4 models show almost 0 signal transducing capabilities. The type-1 chain model, which contains only positive regulations and resembles the structure of a MAPK pathway, shows some signal transducing capabilities. Such systems show ultrasensitive behavior [38], thus filtering out noise, but also turning signal transduction into an analog-digital converter.

## Comparison of coupled feed-forward loops

The 33 coupled multi-input FFLs (Fig 1C) and 26 multi-intermediate FFLs (Fig 1D) show various input-output correlations when tested in the same way as simple FFLs (S3 and S4 Figs). The noise reduction capabilities of these networks were also measured by the coefficient of variation in percentage (%CV) and plotted against the slopes of the input-output correlations as a proxy of signal transduction capacity (Fig 3). Interestingly, networks with better noise reduction usually show better signal transduction as well–there is a general negative correlation between coefficient of variation and slope (Fig 3). The best performing networks are coherent

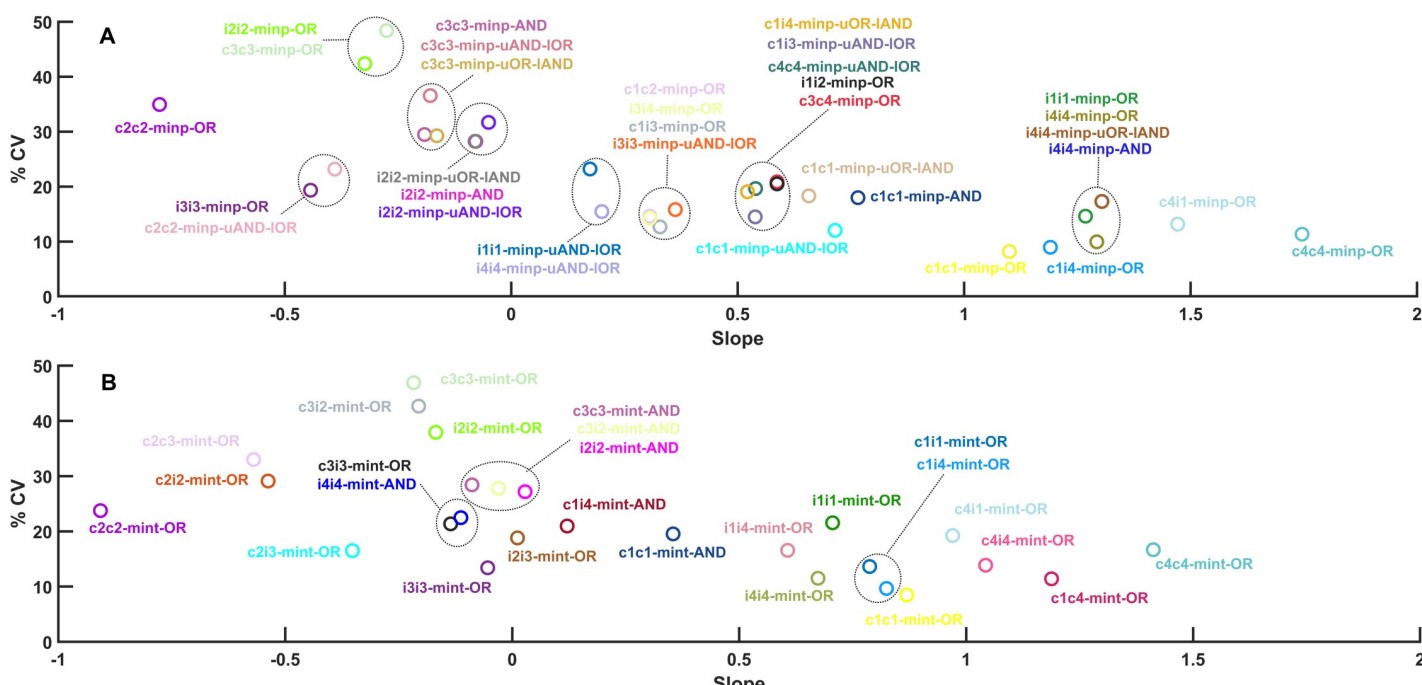

**Fig 3. Signal transducing and noise reducing capabilities of coupled feed-forward loop network motifs.** The coefficient of variation in percentage (% CV) of each combined FFLs was calculated at input = 6 and plotted against the slopes of the input-output relations of S3 and S4 Figs to show how each multi-input coupled FFL (A) and multi-intermediate FFL (B) network performs in noise reduction and signal transduction. All simulations were run with one-step post-translational modification models and the fixed parameter sets presented in the S1 Text.

multi-input FFLs. The best noise reducer is the system where all reactions activate their targets, and the two branches can work independently (c1c1-minp-OR). The best signal transduction we see at a similar coherent, multi-input FFL, but here the long arm of the FFL contains two inverting inhibitory reactions (c4c4-minp-OR). We also observe that coupling between two FFLs at the input level gives better noise reduction than coupling between the same FFLs at the intermediate level (compare networks with similar initial letters in Fig 3A and 3B).

By comparing Fig 2D with the two panels of Fig 3, we can see that coupled FFLs have better noise reducing capabilities than simple FFLs. To compare the best performing simple and combined FFL networks (Fig 4A) in more detail, we have plotted the mean values, highlighted the standard deviations, and labeled the coefficient of variations in percentage (% CV) and slopes of the outputs of each of these networks (Fig 4B–4J). Here, the standard input changes (Fig 1F) were applied to all networks and the outputs show that c1c1-minp-FFL-OR is the best noise reducer (lowest % CV)) and c4c4-minp-OR is the best signal transducer (highest slope value). The basic incoherent 4 FFL also performs quite well, and its various combinations with coherent 1 FFL lead to the best performing combined FFL where the two FFLs are different. We can notice that large negative slopes indicate good inverted signal transduction. FFLs including c2 coherent FFL not only have good, inverted signal transduction but also reasonable noise reduction capabilities.

So far, we have considered a single step modification reaction between active and inactive forms at all levels of the considered FFL networks. Posttranslational signal transducing

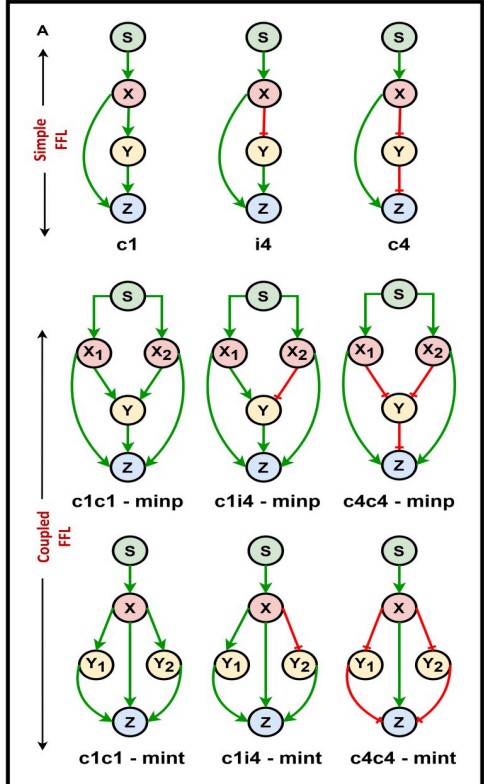

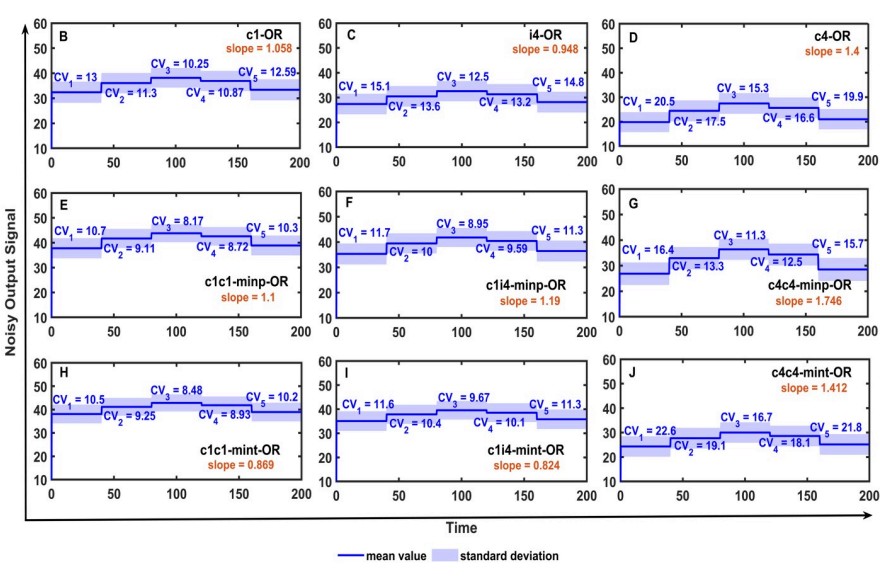

**Fig 4. Comparison of noise reducing and signal transducing performance of the best performing FFL motifs.** The best performing simple and combined FFL networks are plotted in (A). The corresponding panels with similar layout (B-J) show the performance of each of these networks for the input change presented in Fig 1F. The mean of the output active forms of Z molecules of panel A are plotted with solid blue lines, ± standard deviation is plotted with shading, and coefficient of variation in percentage (% CV) values and slopes of input-output relation curves are labeled on all plots. All simulations were run with one-step post-translational modification models and the fixed parameter sets presented in the S1 Text.

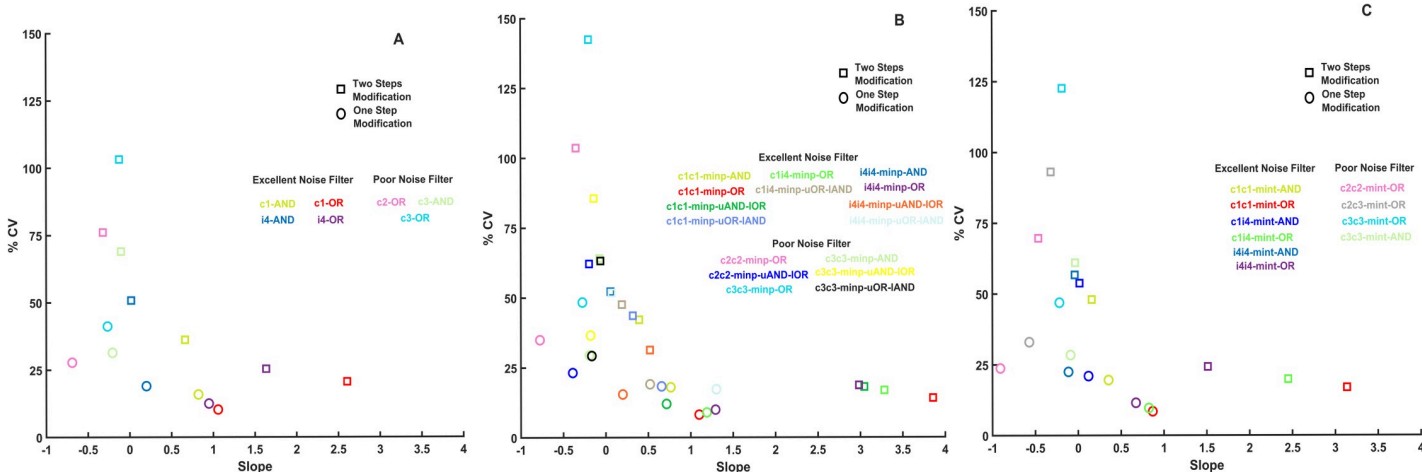

**Fig 5. Comparison between FFLs containing one step and two steps modifications at each regulatory layer.** Input-output relation slope and coefficient of variation in percentage (% CV) data from Figs 2D and 3 are plotted together with similar data from systems where at each layer, molecules go through 2 modification steps before getting activated. Simple FFLs (A), multi-input (B) and multi-intermediate (C) coupled FFLs all show that two step modification systems have a higher % CV. All simulations were run with the fixed parameter sets presented in the S1 Text.

networks often function with multisite modifications, which can lead to highly non-linear behavior [39,40]. To test if multisite modifications can improve the performance of FFLs, we have created a version of the model where at each layer, molecules go through two modification steps, which are both induced by the same activator. The comparison of input-output relation slopes and coefficient of variation in percentage (% CV) produced by these networks and similar networks with single modifications shows that multisite modification increases both % CV and signal transduction (Fig 5). Thus, multisite modification does not help the noise reduction capabilities of FFLs, but it does increase the signal transducing capabilities of FFL motifs.

On the same plots, we can notice that coupled FFLs with AND gates tend to have a worse noise reducing capability than networks with OR gates (Fig 5). On the busier figure Fig 3, it was harder to notice this trend, but the same relationship holds for all networks and especially when comparing good and bad noise reducers (Fig 6). Through parallel signaling in coupled

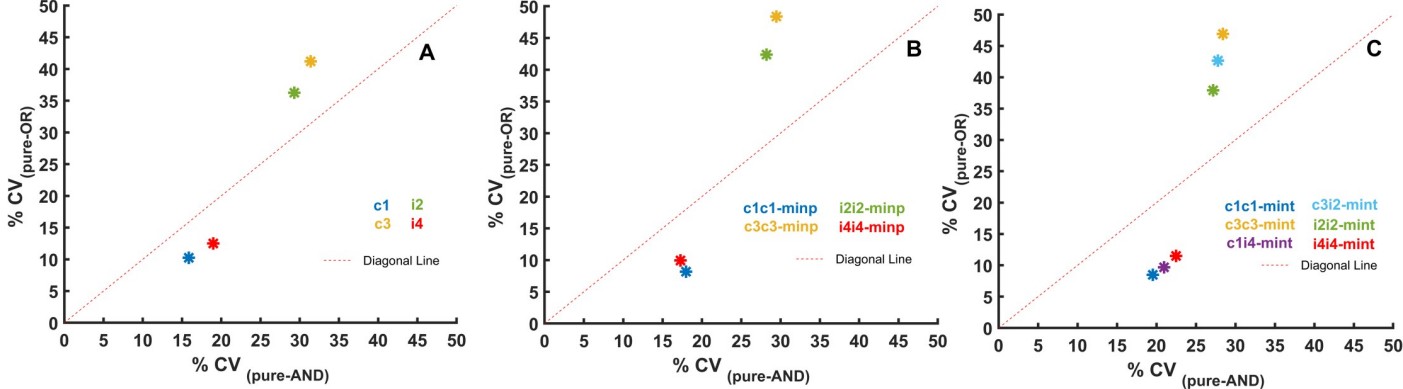

**Fig 6. Comparison between the noise reducing capabilities of FFLs containing AND and OR logic gates.** Data from Figs 2D and 3 are displayed to show how the coefficient of variation (% CV) of AND gate containing networks compare to OR gate driven networks. OR gate containing networks show a lower CV% than AND gate networks (are below the diagonal) and this finding holds true for simple FFLs (A), multi-input (B), and multi-intermediate (C) coupled FFLs. All FFL combinations where both OR and AND gates are possible are plotted. The fixed parameter sets given in the S1 Text were used in all simulations.

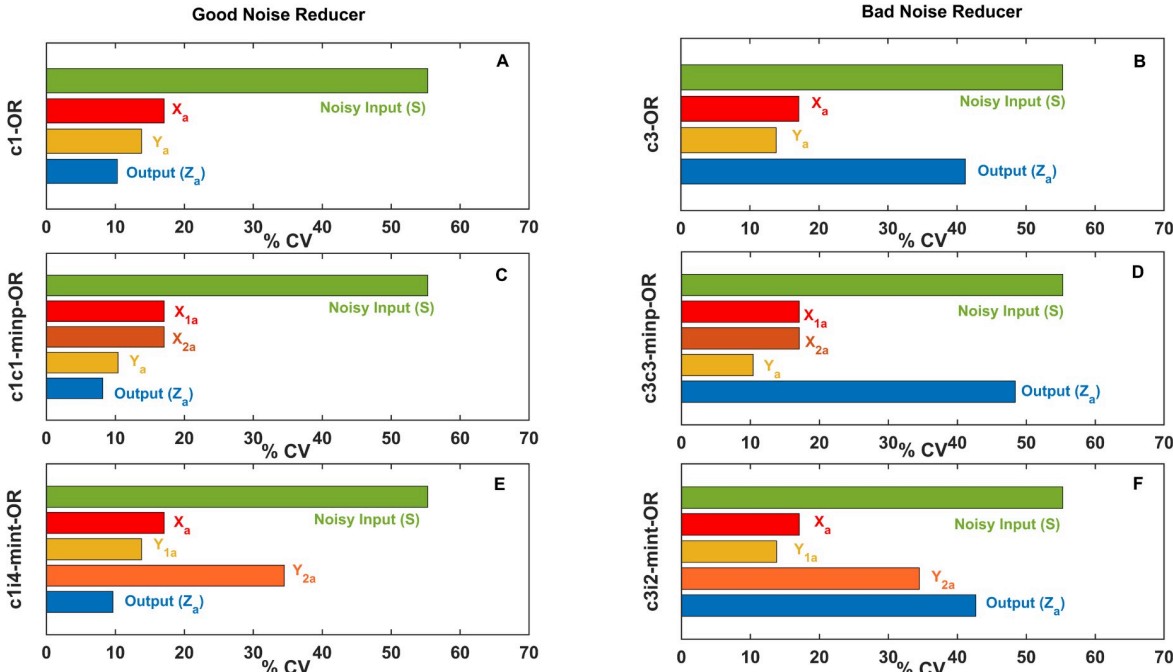

**Fig 7. Noise reduction at each layers of best and worst performing FFLs.** Coefficient of variations (% CV) of each species of best noise reducing (A, C, E)and worst noise reducing (B, D, F) FFLs of different types are plotted for mean input level = 6. Simple FFLs: c1-OR (A), c3-OR (B), multi-input FFLs: c1c1-minp-OR (C), c3c3-minp-OR (D) and multi-intermediate FFLs: c1i4-mint-OR (E), c3i2-mint-OR (subplot F) were investigated here.

FFLs, the noise in the input signal travels down to lower layers of the network and these correlated noises enhance their joint effect when they meet at the output layer with an AND gate. If they meet at an OR gate, this increase in noise seems to be reduced.

## Discussion

Although biological systems are noisy, they are resilient and make reliable decisions when integrating environmental information [41]. Signaling pathways provide rapid response to ligands binding to extracellular receptors by initiating posttranslational modifications on downstream molecules. MAPK pathways have been heavily investigated by mathematical models [38,42,43] and we have learned a lot about their dynamical behavior and signal transducing capabilities. At the same time, feed-forward loop network motifs were investigated by similar approaches to learn about their noise reducing and adaptive behaviors [11,44,45]. Here we have combined these two concepts to study how coupled three-layer posttranslational signaling units can perform noise reduction and signal transduction.

Our systematic study of all possible combinations of FFLs identified that coupled FFLs are better noise reducers and better signal transducers than corresponding single FFLs. This is especially true when OR gates are functioning in multi-input steps of the pathways. Posttranslational modifications often happen in multiple steps [38,40]. We have shown that such highly non-linear transitions make FFL-based signal transduction noisier, but at the same time, this can improve their signal transducing capabilities.

A few interesting features emerge from our analysis: The FFLs which show the best signal transduction capacity all contain an activatory direct arm from input to output (Figs 2D and 3). This means that node $X$ should activate node $Z$ directly for good signal transduction. It can

also be observed that good signal transduction can be observed when at least one copy of the intermediate molecule $Y$ is inhibited by the input node $X$ and the output gates follow an OR logic. These all hold for the FFLs c4c4-minp-OR, c4i1-minp-OR, c4c4-mint-OR, c1i4-mint-OR, c4-OR, which have the highest signal transducing capacity. On the other hand, efficient noise reduction we see in FFLs with a similar activatory direct arm between $X$ and $Z$, but here this is combined with a reaction, where $Y$ activates $Z$ (Figs 2D and 3) and again, multi-input steps follow an OR logic. Following this description, we found c1c1-minp-OR, c1i4-minp-OR, c1c1-mint-OR, c1i4-mint-OR, c1-OR, i4-OR to be the best noise reducer FFLs.

We were speculating what causes the good noise reducing capabilities of some FFLs, while others perform bad in this task. When we perturb the input with smaller or larger noise, we did not see major differences in the noise level at the output layer (S7 Fig), showing that the noise at each layer of posttranslational FFLs is their inherent feature and quite independent on the noise at the input layer. This finding holds for all layers of the networks (S7 Fig). Interestingly, plots of noisiness (% CV) in each layer of FFLs shows great difference between good and bad noise reducing FFLs (Fig 7). In good noise reducers the noise is getting decreased as we go to lower layers of FFLs (with little twists in incoherent FFLs), while in bad noise reducers often the middle layer (species $Y$) is the best noise reduced. This finding highlights that networks we claimed as bad noise reducers, could be still observed in biology, when important signals are used for cross-talking from middle layers of pathways [46]. We can also conclude from this, that good noise reducing networks at the output layers are efficient in this as noise keep reducing as signal progresses in lower layers of FFLs.

Our study has several limitations. We considered the total abundance of proteins at each layer to be equal, whereas for better signal amplification, the abundance increases at each layer [42]. We also limited the analysis to the coupling of two FFLs, whereas signaling pathways have much wider crosstalk [47]. Despite these, and possible other limitations, our systematic analysis provides a thorough characterization of the noise reducing and signal processing capabilities of combined FFLs. The study highlights that specially coupled FFLs of the coherent1 and 4 and the incoherent 4 types with OR logic gates can serve as good signal transducers and good noise reducers. We believe that the engineering of such coupled FFL network motifs could bring new opportunities to better control the noise and signal transduction in synthetic biological systems [48].

## Supporting information

**S1 Fig. 47 possible isolated FFL and combined FFL networks.** (A) Isolated feed-forward loops: The diagram represents the purely coherent (upper panel) and purely incoherent (lower panel) types of isolated feed-forward loops. Here, the possibilities of AND/OR types of logical gates are taken into consideration. Twelve different kinds of logical connectivity are possible. Coherent and incoherent are represented by the letters 'c' and 'i' respectively. The number next to the letter 'c'/ 'i' designates the type of model (for example, c1 –coherent type 1, i4 – incoherent type 4 model). Activation and inhibition processes are shown by green and red arrow heads, respectively. The noisy input signal (S) regulates $X_a$, which in turn influences $Y_a$ and the output signal ($Z_a$) through the direct and indirect arms of the network. (B) Chain models: The diagram shows all the possible types of chain model. The noisy input signal (S) regulates $X_a$, which in turn influences $Y_a$, and $Y_a$ influences the output signal ($Z_a$). (C) Multi-input coupled feed-forward loops: The diagram shows all the multi-input coupled feed-forward loops (minp-FFL). These networks are subcategorized into purely coherent (left panel), mixture of coherent and incoherent (middle panel) and purely incoherent (right panel) types. The input signal (S) jointly activates two nodes, $X_{1a}$ and $X_{2a}$. These $X_{1a}$, and $X_{2a}$ influence $Y_a$

and $Z_a$ through direct and indirect regulated arms, where $Z_a$ represents the output signal. The green arrows represent activation, and the red arrows represent inhibition. Depending on the network architecture, fully AND, fully OR, upper-AND-lower-OR (uAND-lOR) and upper-OR-lower-AND (uOR-lAND) types of logical gates can be considered. (D) Multi-intermediate coupled feed-forward loops: The diagram represents various types of multi-intermediate coupled feed-forward loop (mint-FFL). These networks are subdivided into purely coherent (left panel), mixture of coherent and incoherent (middle panel) and purely incoherent (right panel) types. The noisy input signal (S) regulates the node $X_a$. $X_a$ acts upon $Y_{1a}$, $Y_{2a}$ and the output signal ($Z_a$) through direct and indirect regulated arms. The green arrows represent activation, and the red arrows represent inhibition. Depending on the network architecture, AND/OR types of logical gates can be constructed.
(TIF)

**S2 Fig. All the possible logic gates for the investigated models.** For the investigated network motifs, all the possible logical gates are shown here. Depending on the network architecture, fully AND, fully OR, upper-AND-lower-OR (uAND-lOR) and upper-OR-lower-AND (uOR-lAND) types of logical gates can be created. (A), (B), (C) and (D) show the types of connectivity for 12 isolated feed-forward loops (FFLs), 4 chain models, 33 multi-input coupled feed-forward loops (minp-FFLs), and 26 multi-intermediate coupled feed-forward loops (mint-FFLs), respectively.
(TIF)

**S3 Fig. Correlation plot for multi-input coupled feed-forward loops.** Correlation between input and output signals for multi-input coupled feed-forward loop models, with single step post translation modification of the species under parameter set $k_1 = k_2 = k_3 = 1$, $k_p = 10$, $k_{pp} = 40$, $k_a = 5$. The slopes obtained by linear regression for each model are given in the legends. Each model is categorized as purely coherent (A), a mixture of coherent-incoherent (B), and purely incoherent (C) types. Results from all networks presented in S2 Fig (AND, OR, upper-AND-lower-OR (uAND-lOR) and upper-OR-lower-AND (uOR-lAND)) are presented in this plot. Based on the signs of the interactions, both negative and positive correlations can be observed. The input was changed in five steps in each model, as presented in Fig 1F, and the outputs were recorded as in Fig 1G.
(TIF)

**S4 Fig. Correlation plot for multi-intermediate coupled feed-forward loops.** Input-output correlation of steady states in multi-intermediate FFLs for purely coherent (A), mixture of coherent-incoherent (B) and purely incoherent (C) networks are plotted. All possible logical connectivity (AND, OR, upper-AND-lower-OR (uAND-lOR) and upper-OR-lower-AND (uOR-lAND)) have been considered. The input was changed in five steps in each model, as presented in Fig 1F, and the outputs were recorded as in Fig 1G. For the calculation, we have considered single-step post-translation modification of the species under parameter set $k_1 = k_2 = k_3 = 1$, $k_p = 10$, $k_{pp} = 40$, $k_a = 5$.
(TIF)

**S5 Fig. Comparison between the estimated noise in one step and two steps post-translational modification models.** The coefficient of variation in percentage (% CV) of the output is plotted as the input to all models was changed as indicated on the abscissa. Results are shown for models with isolated FFL (A), multi-input coupled FFL (B) and multi-intermediate coupled FFL (C) under the parameter set $k_1 = k_2 = k_3 = 1$, $k_p = 10$, $k_{pp} = 40$, $k_a = 5$. In each of these panels, we compared the results between one step (solid line with a circle shaped marker) and two steps (solid line with a square shaped marker) post translational modification. Each model is

represented by a unique color, varying only in the shape of the markers. We can infer that, among all the models shown in this figure, c1c1-minp-OR under one step modification, is a better noise reducer than the c1c1-minp-OR model under two step modification (B). Also, it can be understood that the OR-type of connectivity is less noisy compared to the AND-type of connectivity. This holds true for isolated-FFL (A) and multi-intermediate coupled-FFL (C) too. Furthermore, it can be stated that the multi-input coupled FFL (c1c1-minp-OR) (B) is the best noise filter compared to the rest (A, C).
(TIFF)

**S6 Fig. Correlation plot for _Chain models_.** We plot correlation between input and output signals of chain models, considered with single step post translation modification of the models, with parameter set $k_1 = k_2 = k_3 = 1$, $k_p = 10$, $k_{pp} = 40$, $k_a = 5$ The caption lists the slope values derived from each model. In each model, the input was modified in five steps as shown in Fig 1F, and the outputs were recorded as shown in Fig 1G.
(TIF)

**S7 Fig. Noise levels in each layer of a good and bad noise reducing models.** By increasing (right hand column) and decreasing (left hand column) the level of noise in the input (by changing the kinetics of _S_ synthesis, see below in the S1 Text.) and comparing with the original noisy input (middle column), we show how noise propagating through the pathways for c1-OR (A, B, C), c1c1-minp-FFL (G, H, I), c3-OR (D, E, F), c3c3-minp-FFL (J, K, L), c1i4-mint-FFL (M, N, O), and c3i2-mint-FFL (P, Q, R) networks at mean input = 6. The level of noise in the output is largely independent of the degree of noise in the input. We consider single step post translation modification of these networks with parameter set $k_1 = k_2 = k_3 = 1$, $k_p = 10$, $k_{pp} = 40$, $k_a = 5$.
(TIFF)

**S1 Text. Model generation and parameterization.**
(DOCX)

**S2 Text. Examples of Kaemika codes of models.**
(DOCX)

## Acknowledgments

We would like to thank Luca Cardelli, Luca Laurenti and Gábor Szederkényi for useful comments on the manuscript.

## Author Contributions

**Conceptualization:** Suchana Chakravarty, Attila Csikász-Nagy.

**Investigation:** Suchana Chakravarty.

**Supervision:** Attila Csikász-Nagy.

**Visualization:** Suchana Chakravarty.

**Writing – original draft:** Suchana Chakravarty, Attila Csikász-Nagy.

**Writing – review & editing:** Suchana Chakravarty, Attila Csikász-Nagy.

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
