## [Decision Letter · Decision Letter 0]

27 Jul 2021

Dear Ms Chakravarty,

Thank you very much for submitting your manuscript "Systematic Analysis of Noise Reduction Properties of Coupled and Isolated Feed-Forward Loops" for consideration at PLOS Computational Biology.

As with all papers reviewed by the journal, your manuscript was reviewed by members of the editorial board and by several independent reviewers. In light of the reviews (below this email), we would like to invite the resubmission of a significantly-revised version that takes into account the reviewers' comments.

We cannot make any decision about publication until we have seen the revised manuscript and your response to the reviewers' comments. Your revised manuscript is also likely to be sent to reviewers for further evaluation.

Sincerely,

Christopher Rao

Associate Editor

PLOS Computational Biology

Jason Haugh

Deputy Editor

PLOS Computational Biology

Reviewer's Responses to Questions

**Comments to the Authors:**

Reviewer #1: This manuscript aims to examine the signal transduction and noise-modulating properties of various signaling network structures comprising feedforward loops. All possible network variants are assembled, varying the type of regulation, dual input function and node duplication. The input signal varies in mean but has a constant Fano factor of 1.83. The mean and Fano factor at the output are compared to those at the input. The study identifies the best-performing network structures for signal transduction and noise reduction.

The manuscript is interesting, timely, well executed and well written. It should be publishable after the following comments are addressed.

(1) The Fano factor measures burstiness (how different a process is from the Poisson process). However, this is not the only measure of noise magnitude. To characterize the standard deviation in terms of the mean, the Coefficient of Variation (CV, defined as standard_deviation/mean) is very widely used. The CV may be preferable for characterizing the quality of signal transduction, whereas the FF is most appropriate to characterize the burstiness of gene expression. For this reason, all the plots showing Fano Factors versus slope should also be made for the CV, using a constant CV at the input.

(2) Keeping the Fano factor constant at the input makes the reader wonder, what would have happened at higher/lower input FF? It would be interesting to explore what happens if the input FF is higher or lower. The same question refers to the input CV.

(3) The manuscript examines signaling (post-translational) interactions. Similar network structures exist in transcriptional regulatory networks. It would be important to discuss how the results would apply to transcriptional regulatory interactions. A relevant reference for FFL clustering may be PMID:15018656.

(4) “…better noise reduction (lower Fano) correlates with better signal transduction (larger slope).” More specifically, this is true only for slope > 0. However, for slope < 0, better noise reduction (lower Fano) seems to correlate with worse signal transduction. Best signal transduction for slope < 0 is near -1, where the FF is the highest. The same is probably true for coupled motifs.

(5) Besides the FFL, negative feedback can effectively reduce noise, as the authors note. In addition to ref. [13], this is also described in PMID:10850721 and PMID:19279212, which may be worth citing.

Reviewer #2: I uploaded the review as an attachment.

Reviewer #3: Review is uploaded as an attachment.

Reviewer #4: In their article entitled “Systematic Analysis of Noise Reduction Properties of Coupled and Isolated Feed- Forward Loops”, Chakravarty & Csikasz-Nagy investigate how nested/coupled feed-forward loop structures of signalling pathways influence noise in signal. They use a tool called Kaemika that calculates variance by linear noise approximation. Although it is known that FFL can reduce noise, the question of how FFL can be combined for better noise reduction is interesting, and will be of interest.

However, the present paper is not well enough developed to really answer these questions, as:

1) The paper lacks a clear materials & methods section, and this it remains unclear what they actually did. I think this would be essential for the paper.

2) From what I understand, they calculate noise using LNA given one set of parameters, and it remains unclear if this result is dependent on the choice of parameters. I suppose that it is very much dependent on the time scales of each reaction, and thus other topologies might perform much better when different time scales are involved in the different layers.

3) The paper provides no intuition why noise reduction occurs. Maybe write down the LNA equations of the best performing networks, then it might be apparent why noise cancels or is reduced, and which parameters are important.

4) It is unclear if the Fano factor is not a good measure of how input noise is reduced, I would argue that the signal/noise ratio is much more important (imagine a network with a lot of input noise, i.e. where only the input noise is important - then the Fano factor would depend on the number of molecules of the output, whereas signal/noise would not). It would be therefore important for the authors to (i) clearly describe the Fano factor and the rational behind using it here.

Minor: The paper needs a thorough proof-read, a re-edit of the figures (fonts very small). The authors should introduce the notation for the FFL-classes more clearly (maybe in an initial figure), so it is immediately clear what e.g. c1i3 is.

**Have the authors made all data and (if applicable) computational code underlying the findings in their manuscript fully available?**

Reviewer #1: Yes

Reviewer #2: Yes

Reviewer #3: Yes

Reviewer #4: **No: **only exemplary scripts are given

PLOS authors have the option to publish the peer review history of their article (what does this mean?). If published, this will include your full peer review and any attached files.

Reviewer #1: No

Reviewer #2: No

Reviewer #3: No

Reviewer #4: No
---

## [Decision Letter · Decision Letter 1]

8 Nov 2021

Dear Ms Chakravarty,

We are pleased to inform you that your manuscript 'Systematic Analysis of Noise Reduction Properties of Coupled and Isolated Feed-Forward Loops' has been provisionally accepted for publication in PLOS Computational Biology.

The reviews of your revised manuscript were mixed: three recommended publication and one did not. As the majority of the reviewers were in favor of publication, we decided to accept your manuscript because we felt that it was a novel contribution to the field.

Best regards,

Christopher Rao

Associate Editor

PLOS Computational Biology

Jason Haugh

Deputy Editor

PLOS Computational Biology

Reviewer's Responses to Questions

**Comments to the Authors:**

Reviewer #1: I would like to thank the Authors for addressing my comments.

I would like to recommend the publication of the revised manuscript.

Reviewer #2: The revision has addressed all of my previous concerns. I recommend the acceptance of the manuscript.

Reviewer #3: The authors have addressed all the points. I, therefore, recommend publication.

The authors may look into the review published in Annu. Rev. Biophys. 2017. 46:131–48. The review presents a nice discussion on mixed feed-forward loop constructions. Citing the review and few lines of discussion would bring in more generality to the authors' work.

Reviewer #4: I unfortunately don't see much improvement.

Although now the authors somewhat varied the parameters, there is no clear analysis on the effects of parameters, nor are there systematic analytical results. It remains an opaque simulation study which does not provide generaliseable insights and where it is unclear how much of the results is general.

On a formal level, again many of the figures have tiny fonts and are very difficult to understand. I also maintain that it is good practice to have a clear description of methods (and not only point to supplementary material).

**Have the authors made all data and (if applicable) computational code underlying the findings in their manuscript fully available?**

Reviewer #1: Yes

Reviewer #2: Yes

Reviewer #3: Yes

Reviewer #4: None

PLOS authors have the option to publish the peer review history of their article (what does this mean?). If published, this will include your full peer review and any attached files.

Reviewer #1: No

Reviewer #2: No

Reviewer #3: No

Reviewer #4: No

---

## [Editor Report · Acceptance letter]

18 Nov 2021

PCOMPBIOL-D-21-00869R1 

Systematic Analysis of Noise Reduction Properties of Coupled and Isolated Feed-Forward Loops

Dear Dr Chakravarty,

I am pleased to inform you that your manuscript has been formally accepted for publication in PLOS Computational Biology. Your manuscript is now with our production department and you will be notified of the publication date in due course.

With kind regards,

Zsanett Szabo
